# Clinical judgment model-based nursing simulation scenario for patients with upper gastrointestinal bleeding: A mixed methods study

AeRi Jang[1], Hyunyoung Park[2]*

1 Department of Nursing, Nambu University, Gwangju, Republic of Korea, 2 College of Nursing, Chonnam National University, Gwangju, Republic of Korea

* hypark@jnu.ac.kr

**Data Availability Statement:** All relevant data are within the paper and its Supporting Information files.

## Abstract

Assisting patients with upper gastrointestinal bleeding is a crucial role for nurses, and as future nurses, students should demonstrate sound clinical judgment. Well-structured, high-quality simulations are useful alternatives to prepare students for clinical practice. However, nursing simulation scenarios focused on enhancing clinical judgment in managing upper gastrointestinal bleeding are limited. This study aims to develop, apply, and evaluate an effective nursing simulation scenario for patients with upper gastrointestinal bleeding based on Tanner's clinical judgment model using a mixed methods study design. A high-fidelity patient simulation was conducted among 80 undergraduate nursing students divided into a simulated control group ($n = 39$) and an experimental group ($n = 41$). Subsequent student performance evaluations used questionnaires and video recordings. After scenario simulations, the students showed a statistically significant increase in theoretical knowledge ($p = 0.001$) and clinical performance skills ($p < 0.001$), but there was no significant increase in self-confidence ($p = 0.291$). According to the video analysis, the "noticing" clinical judgment phase was the most frequently observed phase, while "reflection" was the least frequently observed phase. Additionally, "education" was the most frequently observed nursing domain, and "anxiety" was the least frequently observed domain. Although further simulation repetitions are required to reinforce students' self-confidence when caring for patients with upper gastrointestinal bleeding, the scenario was deemed effective. Moreover, emphasis should be placed on developing various scenarios to strengthen students' clinical judgment skills, especially "reflecting" and "emotional care."

## Introduction

Nursing academic institutions have difficulty securing practice facilities for their students' clinical placements because of the COVID-19 pandemic. As such, simulations have been actively used worldwide as an alternative format for clinical practice [1]. Understanding the clinical judgment process and how nurses think in actual clinical situations are essential

**Funding:** This work was supported by the National Research Foundation of Korea (NRF 2017R1C1B5017463).

**Competing interests:** The authors have declared that no competing interests exist.

considerations for simulations to be appropriately used as an alternative to clinical practice. Therefore, simulations should reinforce practicing clinical judgment by allowing students to think like a nurse. Tanner [2] presented the clinical judgment process as a model that incorporates various tasks, such as noticing, interpreting, responding, and reflecting. Instructors are challenged to provide effective simulations that improve competencies, such as clinical judgment, and prepare students to become future nurses, but these skills only develop over time through experience [3]. Simulations have already been identified as a potent approach for developing nursing students' clinical judgment [4, 5].

Furthermore, well-structured, high-quality simulations have been suggested as effective teaching modalities comparable to hospital-based clinical experiences [6]. However, to effectively improve nursing student competencies using simulations, medical school faculties need to select diverse scenarios reflecting real-world situations through multiple simulation experiences [4, 7]. Despite the COVID-19 pandemic, it is a critical time for educators to share ideas about using simulations to prepare students for clinical environments [1]. Developing simulation scenarios based on the clinical judgment model and evaluating students' clinical judgment experience during the simulations will help construct meaningful simulation experiences that could replace clinical practice during and after the pandemic.

## Simulation scenario development for nursing education

Although upper gastrointestinal bleeding (UGIB) is curable with medications or endoscopic hemostasis treatments, its hospital mortality rate can go as high as 8.7% [8, 9]. As such, patients with UGIB require rapid and timely diagnosis and treatment. To date, published guidelines have emphasized the nurses' role in interpreting signs, symptoms, and risk factors related to UGIB [10]. Adding specific materials to nursing students' curricula to help them rapidly develop their capabilities and display their effectiveness in caring for patients with UGIB is essential to reinforce their knowledge.

Nursing educators encourage their students to develop clinical judgment and apply their knowledge and experience in decision-making or patient care [2, 11]. Specifically, Tanner [2] proposed the clinical judgment model (CJM), which explains how nurses should think in practical situations where clinical judgment is required, and includes four phases: noticing, interpreting, responding, and reflecting. The most effective proposed teaching method for enhancing nursing students' clinical judgment is simulation-based learning [12], as it allows students to acquire the required knowledge, skills, and attitudes in a practical manner [13].

Although several simulation-related studies have used Tanner's model, such research focused on post-simulation debriefing or assessment rubric development [14–16]. However, improving the nurses' clinical judgment requires a simulation scenario based on clinical judgment [17]. In this area, studies based on Tanner's model are lacking.

Notably, in previous literature related to simulations for patients with UGIB, clinical judgment focused on scenario development and the student self-evaluation processes [18]. As such, these scenarios' effects have only been evaluated based on self-confidence and satisfaction levels [19]. Other scenarios were also developed to improve the endoscopic techniques of resident physicians [20].

## Purpose

The study has two major purposes. First, this study aims to develop a simulation-based learning scenario integrated with the phases of Tanner's model to create simulations that effectively improve nursing students' capability to advance in the clinical field. Second, this study applies and evaluates a scenario's effectiveness when caring for patients with UGIB.

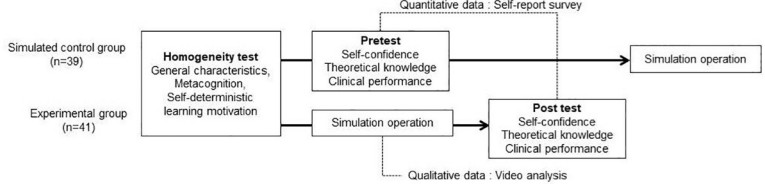

**Fig 1. Study process.**

## Materials and methods

### Design

The study utilized a mixed methods design (Fig 1). Specifically, a self-report survey and video analysis were used to evaluate the developed scenario's effectiveness and review the experimental group's simulations, respectively [21].

### Participants

Recruitment was announced on a notice board at Songwon University, and the study's participants mainly included 3rd-year nursing students who took a course on the digestive system. The students recruited for the study understood its purpose and provided their written consent. Among the recruited students, those who had difficulties participating in discussions or team activities arising from physical or mental issues, disagreed with the confidentiality agreement related to the scenario's operation, or took pictures and video recordings during the study were excluded. The study's results were obtained from 80 participants (4 dropped out for personal reasons) divided into two groups: 41 students in the experimental group and 39 students in the simulated control group. To calculate the number of the study participants, G-power 3.19.2 and "ANOVA: repeated measures, within-between interaction" were chosen (effect size = 0.25, $\alpha$ = 0.05, power = 0.95, correlation coefficient = 0.5). The experimental and control groups included 36 students each, which was considered representative of the larger student population.

### Scenario development

The high-fidelity human patient simulation scenario was developed by two adult nursing professors and one simulation professor for 3rd-year medical school students in South Korea. This scenario mainly focused on applying the clinical judgment steps (assessing, diagnosing, planning, implementing, and evaluating) in the nursing process to provide urgent care for patients with UGIB. The title, learning objectives, simulation operating methods, and teaching materials were determined to develop the scenario, and available facilities and equipment were identified. In line with the learning objectives, five domains required sound clinical judgment for caring for patients with UGIB, including bleeding, pain, nutrition, anxiety, and education. The scenario was subsequently developed to allow students to practice the four phases of Tanner's CJM in each of the five domains. Content related to medical treatment was based on guidelines provided by Bai and Li [10]. In the developed scenario, a 48-year-old female with intermittent epigastric pain visited the hospital for further evaluation because of hematemesis, and her primary diagnosis was gastric ulcer bleeding. The scenario presented a situation where the patient showed vomited blood in a paper cup to the nurse during a regular ward round. Students were assigned roles in each simulation team, including one charge nurse, two or three acting nurses, and one documenting nurse.

Two nurses with over five years of experience in a gastroenterology ward confirmed the clinical practice's reproducibility to verify the developed scenario's validity. One gastroenterologist confirmed the medical aspect of the scenario, and two nursing professors in charge of the simulations confirmed the feasibility of the simulated operation. The initial scenario was revised based on feedback from the expert panel and finalized after the pilot test.

**Study assistants' training.** The simulation operator and instructor have over two years of clinical experience and simulation operation experience. Both underwent two rounds of 2 h training sessions on study-related information conducted by a researcher. The instructor also participated in an 8 h simulation-related education session. During the training period, the data collector was taught the questionnaire completion method and common cautions, such as confidentiality and data management.

**Measurement tools.** For homogeneity tests between the experimental and simulated control groups, the study investigated metacognition [22], self-deterministic learning motivation [23], and critical thinking ability [24], each of which can affect the clinical judgment process using questionnaires (S1 and S2 Files).

This study adopted a strategy proposed by Yang [25] as a self-regulated academic measuring tool modified to fit nursing students using a Cronbach's alpha coefficient of 0.910 to measure nursing students' metacognition. For self-deterministic learning motivation, the academic self-regulation questionnaire (SRQ-A) was adapted and validated, following the method of Bak et al. [26], using a Cronbach's alpha coefficient of 0.900. The study also used the thinking tendency measuring tool for Yoon's [27] critical thinking using a Cronbach's alpha coefficient of 0.901.

This study gauged the self-confidence, theoretical knowledge, and clinical performance skills to verify the scenario's effectiveness based on Tanner's CJM. Jeffries et al. [28] suggested the participants' responses and learned behavior to measure learning achievement in simulation education. The corresponding tool developed here was verified by two adult nursing professors and two clinical nurses, and the content validity indexes of all comprised items were over 1.0. For self-confidence measurements, the instrument consisted of seven items, based on a Cronbach's alpha coefficient of 0.933. Meanwhile, the theoretical knowledge instrument consisted of 10 items, with higher scores showing a higher level of knowledge. As for clinical performance skills, the instrument consisted of 27 items with a Cronbach's alpha coefficient of 0.931. Higher scores on the scale represented a higher level of the item being measured.

**Data collection.** Data collection was conducted from November 23 to December 1, 2018, at the Songwon University simulation lab through a survey using structured self-report questionnaires and an observation method using video analysis. One week before the simulation operation, the experimental group performed self-learning activities as prelearning using online materials such as video clips. The materials developed covered numerous aspects, including theoretical knowledge (30 min) and performance skill (30 min), related to nursing care for patients with UGIB. Each student team consisted of four to five members, and a total of nine teams were involved in the simulation.

The simulation class lasted for 100 min, which was comprised of a presimulation team activity (60 min), simulation operation (20 min), and postsimulation (20 min). The simulation operator played the doctor and monitored the simulation process, while the simulator moderator played the patient and adjusted the simulation. For the simulated control group, general characteristics and homogeneity-related variables were previously measured. A pretest to determine confidence, theoretical knowledge, and clinical performance skill levels was immediately provided after prelearning using online materials. The pretest for the simulated control group and the posttest for the experimental group were conducted at one-week intervals to prevent treatment diffusion. A separate surveyor conducted the data survey.

The entire simulation operation process for the nine teams in the experimental group was recorded using a high-definition camcorder for video analysis. The analysis was conducted by an expert with over four years of nursing experience in a gastroenterology ward and an expert with over three years of simulation operating experience. The video analysis team first classified the nursing actions expected for each clinical judgment process in the scenario developed before proceeding to the video analysis. Both experts independently examined the frequency of students' clinical judgment behaviors by reviewing and analyzing the recorded videos.

In this study, the students' clinical judgment behavior is defined as the actual performance of actions for noticing, interpreting, responding, and reflecting in the nursing care domains of bleeding, pain, nutrition, anxiety, and education involving patients with UGIB. The examiners' rating reliability was substantial, with a Cohen's kappa coefficient ($\kappa$) of 0.7. Any disagreement between the examiners was resolved by consensus.

## Data analysis

Collected data were analyzed using Statistical Package for the Social Sciences (SPSS Version 20.0; IBM, 2012). The participants' general characteristics and the value of each variable were calculated using descriptive statistics, including frequency, percentage, mean, and standard deviation. The homogeneity test of participants was performed using a Chi-square test and a t-test. Verification of the effectiveness of the scenario application was analyzed using a t-test. The video analysis results on the frequency of students' clinical judgment were examined based on the mean, minimum, and maximum.

## Ethical considerations

Ethical approval was obtained from the Ministry of Health and Welfare–designated Korean Public Institutional Review Board (IRB No. PO1-201811-13-004). A research assistant obtained written informed consent from each participant after explaining this study's purpose, the lack of disadvantages in refusal, and the possibility of withdrawal at any time. For the control group, the same simulation education was provided after the experimental group's posttest.

## Results

### Developed scenario

The scenario's final version allowed nursing students to experience Tanner's clinical judgment process in the five areas (bleeding, pain, nutrition, anxiety, and education) required in the nursing and care of patients with UGIB. For instance, in the "noticing" stage under the "bleeding" domain, 10 cues (including blood pressure decrease and heart rate increase) were provided in which the participants could make clinical judgments. In the "interpreting and responding" stage, 17 items, including vital sign check or oxygen saturation level check, were established as appropriate actions. In the "reflecting" stage, three items, including checking hemoglobin test results after blood transfusions, were established. This detailed scenario is further described in Table 1.

### General characteristics and homogeneity test

There were 31 female students (75.6%) and 10 male students (24.4%) in the experimental group, and 35 female students (89.7%) and 4 male students (10.3%) in the control group. The average ages of the experimental and control groups were 23.10 and 23.05, respectively. According to the homogeneity test, neither group displayed statistically significant differences

**Table 1. Nursing scenario for patients with upper gastrointestinal bleeding.**

| Clinical Judgment | Noticing | | Interpreting and Responding | Reflecting |
|---|---|---|---|---|
| **Bleeding** | **Setting:** | | Check vital signs, saturation | Recheck Hgb in the laboratory after transfusion |
| | BP[a]: Dropped from 110/70 to 90/60 | | | |
| | HR[b]: Rise from 112 to 124 bpm | | Check for hematemesis, hemoptysis | Check hematemesis characteristics |
| | | | Check bowel sounds | |
| | Bowel sound: Hyperactive | | Check general laboratory Hgb | |
| | 30 cc of hematemesis blood in a cup on the patient table | | Check past medication | Check V/S[i] |
| | | | Check hepatitis history | Check EGD[j] result |
| | | | Perform digital rectal exam | |
| | Hgb[c] on general blood laboratory test: fell from 10.6* to 8.9 g/dl (*upon hospitalization) | | Check for nausea/vomiting | |
| | | | Remained fluid rate elevation | |
| | | | Apply head-up position | |
| | Hgb on additional laboratory: 7.6 | | Perform pretransfusion test | |
| | Ibuprofen 200 mg 2T PRN[d] when having abdominal pain | | Perform transfusion | |
| | | | Start Pantoloc 5 ample mixed fluid injection | |
| | **Patient symptoms:** | | If needed, allow low $O_2$[h] inhalation | |
| | "Blood comes up through the mouth." | | Put EGD on standby | |
| | "I threw up in the cup just in case." | | | |
| | "Heartbeat seems to be getting fast, and I feel like I'm out of breath." | | Start NPO[l] (abstain from food) | |
| | | | Require bed rest | |
| | | | | |
| **Pain** | **Setting:** | | Check abdominal pain (PQRST) | Reassess pain |
| | Ibuprofen 200 mg 2 tablet PRN when experiencing abdominal pain | | Inject with painkiller following doctor's prescription | |
| | Constant epigastric pain noted on hospital records | | | |
| | **Patient symptoms:** | | | |
| | PQRST[e] pain assessment: LUQ[f], sharp pain, 3 times a day after meals; pain lasts approx. 30 min; NRS[g] grade 6; no radiating pain | | | |
| **Nutrition** | **Setting:** | | Check nutrition state on additional albumin, total protein blood test | Check weight change |
| | Height/weight = 162 cm / 68 kg upon hospitalization | | | Conduct postcheck on the nutrition-related blood test |
| | Body Weight fell to 66 kg 2 days after hospitalization | | Check skin tension and elasticity | |
| | Additional laboratory tests: Total protein 5.5 g/dl, albumin 3.2 g/dl | | Check weight change | |
| | **Patient symptoms:** | | Start peripheral TPN[k] | |
| | "I feel weak all over." | | | |
| | "I barely ate." | | | |
| | "I feel dizzy sometimes. . ." | | | |
| | "I am wet all over like I have cold sweats." | | | |
| | "I lost 2 kg after hospitalization. (I usually weigh 68 kg, but it went down to 66 kg when I checked yesterday.)" | | | |
| **Anxiety** | **Patient symptoms:** | | Check anxiety | Reassess patient's anxiety |
| | "What's happening with me?" | | Stay with patient | |
| | "Isn't this a serious problem?" | | Explain treatment procedures to patient | |
| | "Do all ulcer patients throw up blood as I do?" | | | |
| | "What if I throw up blood when I get back home?" | | | |

(*Continued*)

**Table 1.** (Continued)

| Clinical Judgment | Noticing | | Interpreting and Responding | Reflecting |
|---|---|---|---|---|
| Education | **Setting:** | | Explain blood test results | Recheck additional inquiries from patient |
| | Social history on hospital records: 2 cups of coffee/day, 2 glasses of soju/week | | Inform patient about disease, complications, medication, exercise, and feeding | |
| | **Patient symptoms:** | | | |
| | "What's happening with me?" | | | |
| | "May I have coffee? I like it." | | | |
| | "Until when should I be banned from food?" | | | |
| | "Aren't all ulcer medications useless? What kinds of medicine should I take?" | | | |

[a]BP = blood pressure

[b]HR = heart rate

[c]Hgb = hemoglobin

[d]PRN = prescribed as needed

[e]PQRST = Personal Questionnaire Rapid Scaling Technique

[f]LUQ = left upper quadrant

[g]NRS = numerical rating scale

[h]$O_2$ = oxygen

[i]V/S = vital sign

[j]EGD = esophago-gastro-duodenoscopy

[k]TPN = total parenteral nutrition

[l]NPO = nothing per oral

in gender, age, satisfaction levels in their nursing majors, adult nursing scores in the previous semester, metacognition, self-deterministic learning motivation, or critical thinking (Table 2).

## Verification of effectiveness on scenario application

Although there was no significant increase in self-confidence ($p = 0.291$), the simulated control group's pretest score was 22.41, and the posttest score of the experimental group was 23.49. Theoretical knowledge showed a statistically significant increase ($p = 0.001$), presenting a pretest score of 3.36 in the control group and a posttest score of 4.71 in the experimental group.

**Table 2.** General characteristics and homogeneity of experimental and control groups ($n = 80$).

| Characteristics | | Cont. ($n = 39$) | Exp. ($n = 41$) | $x^2$ or t | $p$ |
|---|---|---|---|---|---|
| | | n (%) or M ± SD | n (%) or M ± SD | | |
| **Sex** | **Female** | 35 (89.7) | 31 (75.6) | 1.69 | 0.096 |
| | **Male** | 4 (10.3) | 10 (24.4) | | |
| **Age (year)** | | 23.05 ± 1.99 | 23.10 ± 2.62 | 0.09 | 0.929 |
| **Satisfaction level in their nursing major** | | 2.83 ± 0.72 | 2.10 ± 0.73 | -1.09 | 0.281 |
| **Adult health nursing scores in the previous semester** | | 2.83 ± 0.72 | 2.96 ± 0.73 | 0.79 | 0.430 |
| **Metacognition** | | 107.90 ± 14.26 | 108.05 ± 13.51 | 0.00 | 0.961 |
| **Self-deterministic learning motivation** | | 42.69 ± 7.02 | 43.02 ± 5.10 | 0.09 | 0.809 |
| **Critical thinking** | | 94.23 ± 12.08 | 96.98 ± 9.69 | 1.12 | 0.268 |

**Table 3. Comparison between experimental and control groups (_n_ = 80).**

| Characteristics | Cont. pretest (_n_ = 39) | Exp. posttest (_n_ = 41) | t (_p_) |
|---|---|---|---|
| | M ± SD | M ± SD | |
| Self-confidence | 22.41 ± 5.04 | 23.49 ± 3.99 | 1.06 (0.291) |
| Knowledge | 3.36 ± 1.58 | 4.71 ± 1.68 | 3.51 (0.001) |
| Clinical performance | 27.44 ± 10.42 | 34.98 ± 8.78 | 3.70 (< 0.001) |

Clinical performance skills also showed a statistically significant increase ($p < 0.001$), presenting a pretest score of 27.44 in the control group and a posttest score of 34.98 in the experimental group (Table 3).

According to the video analysis, students performed the "noticing" stage 12.7 times; the "interpreting and responding" stage was performed 9.4 times, and the "reflecting" stage was performed 9.0 times. Based on the nursing domain results, the domain with the highest observed frequency compared to the expected frequency was "education" (expected = 8, observed = 5.4), and the lowest was "anxiety" (expected = 8, observed = 0.4). As observed in Tanner's CJM phases, the phase with the highest observed frequency compared to the expected frequency was "noticing" (expected = 30. observed = 12.7), and the lowest was "reflecting" (expected = 9, observed = 2.1; Table 4).

## Discussion

This study's UGIB patient scenario was developed based on Tanner's clinical judgment process for five nursing domains. Existing research on simulation-based education for nursing students involving patients with UGIB generally focused on training and evaluating nursing skills required for gastrointestinal emergencies rather than clinical judgment—the focus of this study [18, 19]. In previous studies, rather than developing a scenario, Tanner's model was used to analyze students' clinical judgment skills after applying the simulation [15] or verifying the effectiveness of a scenario based on objectified figures of clinical judgment skill improvements [14, 16]. However, the scenario developed in this study went beyond the evaluation of clinical skills when caring for patients with UGIB. It was a systemically designed scenario appropriate for nursing care to promote clinical judgment in the "bleeding," "pain," "nutrition," "anxiety," and "education" nursing domains, as required in clinical settings for patients with UGIB. This scenario is significant because it is the first trial that requires nursing students to think like real nurses in the field through a simulation that is faithful to the clinical judgment process.

The results showed a statistically significant increase in theoretical knowledge and clinical performance skills; self-confidence also increased, although it was not statistically significant. A previous meta-analysis study on simulation effects, knowledge, and performance showed a statistically significant increase, and self-confidence showed a statistically significant increase due to simulation learning effects [18, 19, 29]. Similarly, Pereia-Salgado et al. [30] operated simulations for "advance care planning" for nurses, where self-confidence results displayed a statistically significant increase in "initiating" and "revisiting." Here, the nursing simulation significantly increased the participants' self-confidence compared to the presimulation.

The findings of previous studies differ from those of the present study for the following reasons. First, because of this study's short-term application of the simulation scenario, it was difficult to attain a statistically significant increase in self-confidence. Moreover, the simulation was likely a burden to the students because it allowed them to learn about the clinical judgment process. Further research should investigate the development and repetitive application of various simulation scenarios that promote students' self-confidence in using clinical judgment in

**Table 4. Results of video analysis.**

| Clinical judgment | | Expected value | Observed value | | | Most frequently observed nursing activities (Examples included below) |
|---|---|---|---|---|---|---|
| | | | Average value | Minimum value | Maximum value | |
| **Total** | **Noticing** | 30 | 12.7 | 6 | 20 | |
| | **Interpreting and responding** | 28 | 9.4 | 5 | 15 | |
| | **Reflecting** | 9 | 2.1 | 0 | 6 | |
| **Bleeding** | **Noticing** | 10 | 6.9 | 4 | 10 | Distinguish hematemesis and hemoptysis |
| | | | | | | Check bowel sounds for hemorrhage status |
| | | | | | | Check BP[a] drop |
| | | | | | | Perform DRE[b] to check actual hemorrhage |
| | **Interpreting and responding** | 17 | 4.6 | 4 | 6 | Apply head-up position |
| | | | | | | Start NPO[c] (abstain from food) |
| | | | | | | Get a prescription for transfusion after reporting to the treating physician |
| | **Reflecting** | 3 | 1 | 0 | 2 | Check V/S[d] after blood transfusion |
| | | | | | | Recheck for hematemesis symptoms |
| **Pain** | **Noticing** | 3 | 1.6 | 1 | 3 | Check pain level and characteristics using measuring tools |
| | | | | | | Check for symptoms of MI[e] |
| | **Interpreting and responding** | 2 | 1 | 0 | 2 | Inject prescribed painkiller after reporting to treating physician |
| | **Reflecting** | 2 | 0.3 | 0 | 1 | Reassess pain using measuring tool |
| **Nutrition** | **Noticing** | 8 | 2.1 | 1 | 3 | Check albumin and total protein during additional laboratory tests |
| | | | | | | Assess nutrition deficiency due to NPO |
| | **Interpreting and responding** | 4 | 0.6 | 0 | 1 | Apply TPN[f] |
| | **Reflecting** | 2 | 0.3 | 0 | 1 | Check for dizziness and hunger |
| **Anxiety** | **Noticing** | 4 | 0 | 0 | 0 | - |
| | **Interpreting and responding** | 3 | 0.3 | 0 | 1 | Do not leave patient unattended |
| | **Reflecting** | 1 | 0.1 | 0 | 1 | Reflect on patient's concerns |
| **Education** | **Noticing** | 5 | 2.1 | 0 | 4 | Listen attentively to the patient's inquiries on the insufficient effect of medication and hematemesis |
| | **Interpreting and responding** | 2 | 2.9 | 1 | 5 | Explain medication for gastric ulcer |
| | | | | | | Explain current injected medication |
| | | | | | | Inform patient about prohibited medication |
| | **Reflecting** | 1 | 0.4 | 0 | 1 | Check for additional inquiries |

[a]BP = blood pressure

[b]DRE = digital rectal exam

[c]NPO = nothing per oral

[d]V/S = vital sign

[e]MI = myocardial infarction

[f]TPN = total parenteral nutrition

nursing practice. During the simulation-based experience, simulation educators and facilitators should also use effective cues to help participants build self-confidence and achieve expected learning outcomes.

The video analysis results indicate that nursing educators who apply the simulation should be more attentive in enhancing nursing students' performance in the "reflecting" phase of

Tanner's model and in "emotional care." Among the clinical judgment process phases, "reflecting" was not observed as frequently as "noticing" or "interpreting and responding." This result indicates that the simulation educator should provide more cues and sufficient time for the nursing students to reach and finish the "reflecting" clinical judgment process in a scenario operation. The "anxiety" domain displayed the lowest observed frequency. Meanwhile, the observed frequency of the "noticing" domain was 0 instead of the expected value of 4, representing how nursing students actively carried out clinical judgment behaviors for physical care involving bleeding, pain, and nutrition.

Conversely, clinical judgment in emotional care, which helps reduce patients' anxiety during the simulation operation, was insufficient. Empathy, which positively affects patients as it plays an essential factor in the therapeutic relationship between nurses and their patients, is an ability that can be developed through simulations [31]. Therefore, further development and operation of simulation scenarios are needed for students to practice varied and holistic nursing-oriented clinical judgment.

This study's results are somewhat difficult to generalize due to several limitations. The participants were recruited from the nursing department of one university. Thus, the study results can be misinterpreted because of extraneous variables from the simulated control group study design. Moreover, the maturation effect cannot be controlled, and the evaluator's subjectivity could affect the video analysis.

## Conclusions

This study successfully developed and applied a simulation scenario for patients with UGIB based on Tanner's CJM. It also identified the effectiveness of simulations and the current level of clinical judgment among undergraduate nursing students. The scenario was developed to allow students to practice the four phases of clinical judgment in five domains while effectively improving their theoretical knowledge and clinical performance when caring for patients with UGIB. Future studies can follow several suggestions. First, a randomized design study using the developed scenario for patients with UGIB should be conducted with students from different educational institutes. Second, diverse simulation-based education using different scenarios should be developed to strengthen "reflecting" in clinical judgment and "emotional care" in nursing situations. Third, a longitudinal study should evaluate clinical judgment ability improvements after repeatedly applying scenarios based on Tanner's CJM. Fourth, future studies should examine non-face-to-face teaching methods and the effects of using Tanner's model-based simulation in a virtual environment. Fifth and last, this study proposed a well-exposed and evaluated clinical simulation scenario for urgent clinical situations. However, future studies can use a randomized study design to evaluate multiple scenarios of the same clinical situation.

## Supporting information

**S1 File. Original research participant presurvey questionnaire in Korean.**
(DOCX)

**S2 File. Translated research participant presurvey questionnaire.**
(DOCX)

## Acknowledgments

The study's authors would like to express their heartfelt thanks to the gastroenterologist, nurses, professors, consultants who assisted in scenario development, the instructors and

operators who assisted during scenario operation, and the participants in this study. This study could not be accomplished without their valuable time and support.

## Author Contributions

**Conceptualization:** AeRi Jang.

**Data curation:** AeRi Jang.

**Formal analysis:** AeRi Jang, Hyunyoung Park.

**Funding acquisition:** AeRi Jang.

**Investigation:** AeRi Jang.

**Methodology:** AeRi Jang.

**Project administration:** AeRi Jang.

**Resources:** AeRi Jang.

**Software:** AeRi Jang.

**Supervision:** Hyunyoung Park.

**Validation:** Hyunyoung Park.

**Visualization:** Hyunyoung Park.

**Writing – original draft:** AeRi Jang, Hyunyoung Park.

**Writing – review & editing:** Hyunyoung Park.

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
