## [Decision Letter · Decision Letter 0]

12 Feb 2021

PONE-D-20-40900

Clinical Judgment Model-based nursing simulation scenario for patients with upper gastrointestinal bleeding: A mixed-method study

PLOS ONE

Dear Dr. Park,

Thank you for submitting your manuscript to PLOS ONE. After careful consideration, we feel that it has merit but does not fully meet PLOS ONE’s publication criteria as it currently stands. Therefore, we invite you to submit a revised version of the manuscript that addresses the points raised during the review process.

We look forward to receiving your revised manuscript.

Kind regards,

César Leal-Costa, Ph. D

Academic Editor

PLOS ONE

Journal Requirements:

2. In your Methods section, please provide additional information about the participant recruitment method and the demographic details of your participants. Please ensure you have provided sufficient details to replicate the analyses such as:

- a description of any inclusion/exclusion criteria that were applied to participant recruitment

- a statement as to whether your sample can be considered representative of a larger population

- a description of how participants were recruited

- descriptions of where participants were recruited and where the research took place.

Reviewers' comments:

Reviewer's Responses to Questions

**Comments to the Author**

1. Is the manuscript technically sound, and do the data support the conclusions?

Reviewer #1: Yes

Reviewer #2: Partly

2. Has the statistical analysis been performed appropriately and rigorously? 

Reviewer #1: Yes

Reviewer #2: Yes

3. Have the authors made all data underlying the findings in their manuscript fully available?

Reviewer #1: Yes

Reviewer #2: Yes

4. Is the manuscript presented in an intelligible fashion and written in standard English?

Reviewer #1: Yes

Reviewer #2: Yes

5. Review Comments to the Author

Reviewer #1: The paper proposes a well-exposed and evaluated clinical simulation scenario for an urgent clinical situation .For subsequent studies it would be proposed to evaluate multiple scenarios of the same clinical situation with a randomised study design.

Reviewer #2: I realize that a great work and time has been devoted to this paper. This is a topic of great significance to nursing students that can affect the learning and therefore, their future work as a nurse. So I appreciate authors examining this topic.

The paper has a lot of strengths but I think that some changes should be recommended.

Abstract:

Please, avoid using abbreviations in the abstract.

Introduction:

I suggest the authors to explain what role exactly have nursing in the approach of UGIB.

Please, write better the aim of the study.

I dont know if the aim was exactly the develop and apply of the scenario. So I think that the main aim is only the evaluation of the scenario effectiveness. So please, discuss this fact.

Methodology:

It is unclear the design of study. When the authors write "mixed methods", do they refer to qualitative and quantitative design?

Results:

What level, year or course were the students?

Please, explain better the sentence in line 219 "noticing 12.7 times..."

Discussion:

Please, add a subheading for the Conclusions.

I suggest the authors to aknowledge at least for the students.

I hope that these recommendations do not discourage the authors and I want to recommend the authors to continue working on this paper.

6. PLOS authors have the option to publish the peer review history of their article (what does this mean?). If published, this will include your full peer review and any attached files.

Reviewer #1: No

Reviewer #2: No

---

## [Author Response · Author response to Decision Letter 0]

22 Mar 2021

Reviewers’ Comments 

1. Is the manuscript technically sound, and do the data support the conclusions?

Reviewer #1: Yes

Reviewer #2: Partly

2. Has the statistical analysis been performed appropriately and rigorously?

Reviewer #1: Yes

Reviewer #2: Yes

3. Have the authors made all data underlying the findings in their manuscript fully available?

Reviewer #1: Yes

Reviewer #2: Yes

4. Is the manuscript presented in an intelligible fashion and written in standard English?

Reviewer #1: Yes

Reviewer #2: Yes

5. Review Comments to the Author

Reviewer #1: The paper proposes a well-exposed and evaluated clinical simulation scenario for an urgent clinical situation. For subsequent studies it would be proposed to evaluate multiple scenarios of the same clinical situation with a randomised study design.

Response: Thank you for your comment. Per your suggestion, the following sentences were added in the Conclusion section. 

“Fifth and last, this study proposed a well-exposed and evaluated clinical simulation scenario for urgent clinical situations. Future studies can utilize a randomized study design to evaluate multiple scenarios of the same clinical situation.” 

Reviewer #2: I realize that a great work and time has been devoted to this paper. This is a topic of great significance to nursing students that can affect the learning and therefore, their future work as a nurse. So I appreciate authors examining this topic.

The paper has a lot of strengths but I think that some changes should be recommended.

Abstract:

Please, avoid using abbreviations in the abstract.

Response: The abstract no longer contains any abbreviations. 

Introduction:

I suggest the authors to explain what role exactly have nursing in the approach of UGIB.

Please, write better the aim of the study.

I dont know if the aim was exactly the develop and apply of the scenario. So I think that the main aim is only the evaluation of the scenario effectiveness. So please, discuss this fact.

Response: In response to the reviewer’s comment, the Introduction, which details the role of nursing patients with UGIB, is further described as follows. 

“Understanding the clinical judgment process and how nurses think in actual clinical situations are essential considerations for simulations to be appropriately used as an alternative to clinical practice. Therefore, simulations should reinforce practicing clinical judgment by allowing students to think like a nurse. Tanner [2] presented the clinical judgment process as a model that incorporates various tasks, such as noticing, interpreting, responding, and reflecting.” 

The following sentences were also added to clarify the study’s purpose.

“The study has two major purposes. First, this study aims to develop a simulation-based learning scenario integrated with the phases of Tanner’s model to create simulations that effectively improve nursing students’ capability to advance in the clinical field. Second, this study applies and evaluates a scenario’s effectiveness when caring for patients with UGIB.”

Furthermore, the effectiveness of the developed scenario is described in the first paragraph of the Discussion section. 

Methodology:

It is unclear the design of study. When the authors write “mixed methods,” do they refer to qualitative and quantitative design?

Response: The study design’s description was revised as follows to clarify the research methodology. 

“The study utilized a mixed methods design (Fig 1). Specifically, a self-report survey and video analysis were used to evaluate the developed scenario’s effectiveness and review the experimental group’s simulations, respectively [22].”

Results:

What level, year or course were the students?

Please, explain better the sentence in line 219 “noticing 12.7 times...”

Response: 

The students’ course, year, and level were described in the Participants, Scenario development, and Verification of the effectiveness of scenario application subsections. 

“Recruitment was announced on a notice board at Songwon University, and the study’s participants mainly included 3rd-year nursing students who took a course on the digestive system. The students recruited for the study understood its purpose and provided their written consent. Among the recruited students, those who had difficulties participating in discussions or team activities arising from physical or mental issues, disagreed with the confidentiality agreement related to the scenario’s operation, or took pictures and video recordings during the study were excluded.”

“The high-fidelity human patient simulation scenario was developed by two adult nursing professors and one simulation professor for 3rd-year medical school students in South Korea. This scenario mainly focused on applying clinical judgment steps (assessing, diagnosing, planning, implementing, and evaluating) in the nursing process to provide urgent care for patients with UGIB.”

“Based on the nursing domain results, the domain with the highest observed frequency compared to the expected frequency was “education” (expected = 8, observed = 5.4), and the lowest was “anxiety” (expected = 8, observed = 0.4). As observed in Tanner’s CJM phases, the phase with the highest observed frequency compared to the expected frequency was “noticing” (expected = 30. observed = 12.7), and the lowest was “reflecting” (expected = 9, observed = 2.1; Table 4).”

Discussion:

Please, add a subheading for the Conclusions.

I suggest the authors to acknowledge at least for the students.

Response: Per your comment, the Discussion section now has a separate Conclusions subheading. The students recruited for the study were also acknowledged in the appropriate section, which now contains the following statement.

“The study’s authors would like to express their heartfelt thanks to the gastroenterologist, nurses, professors, consultants who assisted in scenario development, the instructors and operators who assisted during scenario operation, and the participants in this study. This study could not be accomplished without their valuable time and support.” 

I hope that these recommendations do not discourage the authors and I want to recommend the authors to continue working on this paper.

6. PLOS authors have the option to publish the peer review history of their article (what does this mean?). If published, this will include your full peer review and any attached files.

Do you want your identity to be public for this peer review? For information about this choice, including consent withdrawal, please see our Privacy Policy.

Reviewer #1: No

Reviewer #2: No

Additionally, the following sections were also revised based on your feedback. 

1. Journal Format: Please ensure that your manuscript meets PLOS ONE’s style requirements, including those for file naming.

Response: The manuscript was revised to meet PLOS ONE’s style requirements, and the files attached with this manuscript are named according to the journal’s prescribed format. 

2. Methods: In your Methods section, please provide additional information about the participant recruitment method and the demographic details of your participants. Please ensure you have provided sufficient details to replicate the analyses such as:

- a description of any inclusion/exclusion criteria that were applied to participant recruitment

- a statement as to whether your sample can be considered representative of a larger population

- a description of how participants were recruited

- descriptions of where participants were recruited and where the research took place.

Response: In response to the reviewer’s comments, the content related to the participants was revised as follows.

“Recruitment was announced on a notice board at Songwon University, and the study’s participants mainly included 3rd-year nursing students who took a course on the digestive system. The students recruited for the study understood its purpose and provided their written consent. Among the recruited students, those who had difficulties participating in discussions or team activities arising from physical or mental issues, disagreed with the confidentiality agreement related to the scenario’s operation, or took pictures and video recordings during the study were excluded.”

“To calculate the number of the study participants, G-power 3.19.2 and “ANOVA: repeated measures, within-between interaction” were chosen (effect size = 0.25, α = 0.05, power = 0.95, correlation coefficient = 0.5). The experimental and control groups included 36 students each, which was considered representative of the larger student population.”

Response: Two files that contain the questionnaires used for the study are included as supplementary files. The first file contains the questionnaires as initially written in Korean. The second file has the same questionnaires translated into English.

S1 File. Original research participant presurvey questionnaire in Korean.

S2 File. Translated research participant presurvey questionnaire.

---

## [Decision Letter · Decision Letter 1]

19 Apr 2021

Clinical judgment model-based nursing simulation scenario for patients with upper gastrointestinal bleeding: A mixed methods study

PONE-D-20-40900R1

Dear Dr. Park,

We’re pleased to inform you that your manuscript has been judged scientifically suitable for publication and will be formally accepted for publication once it meets all outstanding technical requirements.

Kind regards,

César Leal-Costa, Ph. D

Academic Editor

PLOS ONE

Additional Editor Comments (optional):

Reviewers' comments:

Reviewer's Responses to Questions

**Comments to the Author**

1. If the authors have adequately addressed your comments raised in a previous round of review and you feel that this manuscript is now acceptable for publication, you may indicate that here to bypass the “Comments to the Author” section, enter your conflict of interest statement in the “Confidential to Editor” section, and submit your "Accept" recommendation.

Reviewer #2: All comments have been addressed

2. Is the manuscript technically sound, and do the data support the conclusions?

Reviewer #2: Yes

3. Has the statistical analysis been performed appropriately and rigorously? 

Reviewer #2: Yes

4. Have the authors made all data underlying the findings in their manuscript fully available?

Reviewer #2: Yes

5. Is the manuscript presented in an intelligible fashion and written in standard English?

Reviewer #2: Yes

6. Review Comments to the Author

Reviewer #2: Dear Authors,

Congratulations on the work done and the changes made.

I think that the paper has improved.

I wish you luck!

7. PLOS authors have the option to publish the peer review history of their article (what does this mean?). If published, this will include your full peer review and any attached files.

Reviewer #2: No

---

## [Editor Report · Acceptance letter]

23 Apr 2021

PONE-D-20-40900R1 

Clinical judgment model-based nursing simulation scenario for patients with upper gastrointestinal bleeding: A mixed methods study 

Dear Dr. Park:

I'm pleased to inform you that your manuscript has been deemed suitable for publication in PLOS ONE. Congratulations! Your manuscript is now with our production department. 

Kind regards, 

on behalf of

Dr. César Leal-Costa 

Academic Editor

PLOS ONE